# A Study on Cross-Shaped Structure of Invar Material Using Cold Wire Laser Fillet Welding (PART I: Feasibility Study for Weldability)

**Du-Song Kim [1,2], Changmin Pyo [3], Jaewoong Kim [3,*], Jisun Kim [3] and Hee-Keun Lee [2]**

[1] Department of Welding and Joining Science Engineering, Chosun University, Gwangju 61452, Korea; dusong@dsme.co.kr

[2] Welding Engineering R&D Department, Daewoo Shipbuilding Marine Engineering, Geoje 53302, Korea; zetlee@dsme.co.kr

[3] Smart Mobility Materials and Components R&D Group, Korea Institute of Industrial Technology, Gwangju 61012, Korea; changmin@kitech.re.kr (C.P.); kimjisun@kitech.re.kr (J.K.)

\* Correspondence: kjw0607@kitech.re.kr; Tel.: +82-62-600-6480

**Abstract:** With the need for eco-friendly energy increasing rapidly due to global environmental issues, there is a rapidly increasing demand for liquefied natural gas (LNG). LNG is liquefied at minus 163 degrees Celsius, and its volume decreases to 1/600, giving it a relatively higher storage and transport efficiency than gaseous natural gas (NG). The material for the tanks that store cryogenic LNG must be a material with high impact toughness at cryogenic temperatures. Invar, which contains 36% nickel and has a very low coefficient of thermal expansion, is used for the membranes and corner structures of LNG cargo holds. The cross-shaped Invar structure used in an LNG cargo hold is manufactured through manual tungsten inert gas (TIG) fillet welding, which causes welding distortion and weldability problems. This study is a feasibility study that aims to reduce welding distortion, increase weldability with welding speed, and reduce the steps in an existing process by half by replacing the existing manufacturing method with automatic fiber laser fillet welding. Laser welding using fiber laser parameters are controlled for 1.5 and 3.0 mm thick Invar materials and weldability is secured through cross-section observation. Then, the optimal welding conditions with top and back beads secured are derived through a trial and error method.

**Keywords:** LNG (liquefied natural gas); Invar; cold wire laser welding; fillet welding; cross-shaped structure

## 1. Introduction

As environmental contamination and global warming issues emerge, the demand for eco-friendly energy is increasing, and natural gas (NG) is becoming popular since it emits a lower amount of pollutants than petroleum [1–3]. However, NG is transported in the form of liquefied natural gas (LNG) after liquefaction because it is produced in limited regions. Therefore, production technology for an LNG Cargo Containment System (CCS), which can store and transport cryogenic LNG at minus 163 degrees Celsius in a liquefied state, is required. Based on this technology, technologies and industries related to LNG storage and processing such as LFS (LNG fueled ship) using LNG as a fuel and LNG bunkering for fuel injection have been continuously developed [3].

LNG is in a cryogenic state of minus 163 degrees Celsius and there is a risk of explosion in the event of leakage, so care should be taken when handling it. SS400 and other general steels cannot be used due to their brittleness below minus 163 degrees Celsius, so metals that do not have low temperature brittleness must be used. The International Code for the Construction & Equipment of

Ships Carrying Liquefied Gases in Bulk (IGC Code) of the International Maritime Organization (IMO) limits tank materials that can store and transport cryogenic cargo to 9% nickel steel, stainless steel (Ex.A240-304L), 36% nickel steel (Invar), Al5083-0, and high manganese steel materials with excellent cryogenic impact toughness [4,5]. The research for the application of those materials in cryogenic condition was performed before [6,7].

Of these, Invar is used in high temperature and cryogenic environments because it not only is strong in terms of cryogenic brittleness but also has a coefficient of thermal expansion (CTE) of $1.2 \times 10^{-6}$ mm/mmK, which is less thermally deformed than other materials (Table 1 [8]). In particular, it is used in the aerospace industry [9], LNG facility parts [10–14], and pipes in cryogenic environments [14,15]. Invar is used in areas where LNG is directly contacted in an LNG facility, and it is also used as a material for LNG transport pipes. If pipes are made with stainless steel, they should have corrugation or some structure for preparation of the thermal shrinkage.

**Table 1.** Mechanical properties of cryogenic materials (at room temperature).

| Materials | CTE (mm/mmK) | Young's Modulus (GPa) | Density (kg/m$^3$) |
|---|---|---|---|
| Invar | $1.2 \times 10^{-6}$ | 148 | 8100 |
| A240-304L | $17.3 \times 10^{-6}$ | 193 | 8000 |
| AL5083-O | $23.8 \times 10^{-6}$ | 71 | 2660 |

Welding is the most commonly used processing technology to manufacture the LNG storage containers or equipment used in the LNG handling process. In welding, a heat source above a certain level of temperature is applied to the filler metal and the base metal to melt both, and they solidify in a short period of time to bond the materials. Welding is widely used in a range of industrial sites. However, welding distortion occurs due to the thermal stress caused by local heating of a material, so control and correction techniques of the material after welding are required. When butt laser welding is applied to thin plate materials of a specific material, welding can be performed by melting the thin plates without using a filler material [16]. However, a filler material can be added to prevent a welding defect if welding penetration beyond a certain level of depth is required. A type of laser welding called laser welding using a fiber laser is already used for materials such as aluminum and stainless steel, and it is being applied to fillet welding in the form of inserting hot or cold wire materials [17–20].

The structure of Invar with "+ (cross shaped)" or "# (number shaped)"in an LNG cargo hold are manufactured by performing manual fillet welding in four fillet welds with partial joint penetration (PJP) through TIG welding. However, when this approach is used, welding distortion occurs due to the characteristics of TIG welding; the weldability will be inconsistent depending on the operator's skills because it is manual welding, and there could be welding defects. For this reason, a significant amount of time and expense is needed for its correction and control. To solve this problem, this study investigated the use of automatic laser welding using a fiber laser using cold wire performed in two fillet welds with complete joint penetration (CJP), with a welding speed of 1 m/min or higher, instead of the current approach of manual TIG welding in four fillet welds.

The main parameters of laser welding in this study include laser power, welding speed, wire feeding speed, working angle, and beam location. This study derived the fillet welding conditions of laser welding using a fiber laser performed in two fillet welds for the Invar structure in an LNG CCS through a case study based on an experiment on the main parameters of laser welding and examination of a welding cross-section.

## 2. Experiment

### 2.1. Specimen for Experiment

There are two thicknesses used for the Invar used in an LNG cargo hold, i.e., 1.5 and 3.0 mm. This study derived the optimal conditions for laser welding using a fiber laser at each thickness. The Invar used in this study contains 36% nickel, and its chemical composition is shown in Table 2.

**Table 2.** Chemical composition of Invar.

| Material | Composition (%) | | | | |
|----------|------|------|------|------|------|
| | Ni | Mo | C | Mn | Fe |
| Invar (ASTM F1684) | 36 | ~0.5 | ~0.1 | ~0.06 | bal. |
| | P | S | Si | Cr | - |
| | ~0.025 | ~0.025 | ~0.35 | ~0.5 | - |

In this study, a laser welding experiment was conducted on a specimen which was fabricated for an experiment on the Invar structure in an LNG cargo hold. As shown in Figure 1a, three plates were used to form the "+" shape. The size of three plates is equally $500 \times 50 \times 1.5$ mm or $500 \times 50 \times 3.0$ mm. Experiments were performed for the thickness of 1.5 and 3.0 mm, respectively. As shown in Figure 1b, a total of two welding passes were performed, once per joint. This is an improvement over TIG welding, which needs a total of four fillet welds, i.e., two fillet welds per joint (Figure 1c).

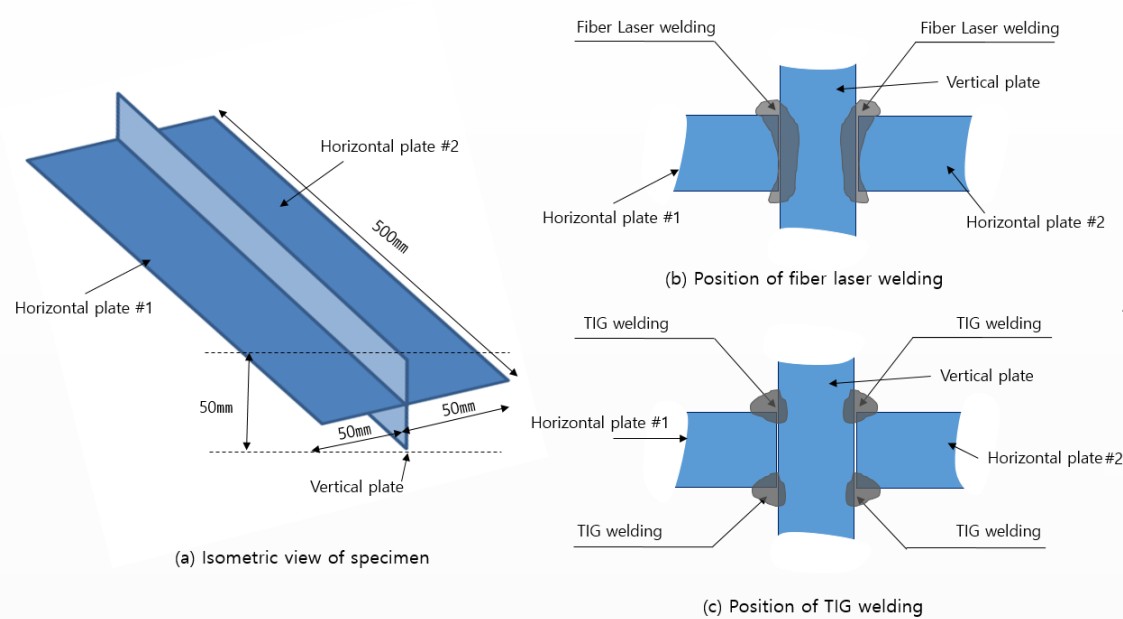

**Figure 1.** Experimental schematic diagram and welding location. (**a**) Isometric view of specimen (**b**) Position of fiber laser welding (**c**) Position of TIG welding.

To control the welding distortion that may occur during laser welding, a jig that can constrain the angular distortion of two horizontal plates was fabricated before the experiment, as shown in Figure 2. Additionally, the jig controls the gap between the base materials and the degree of gap was almost 0 mm in this study.

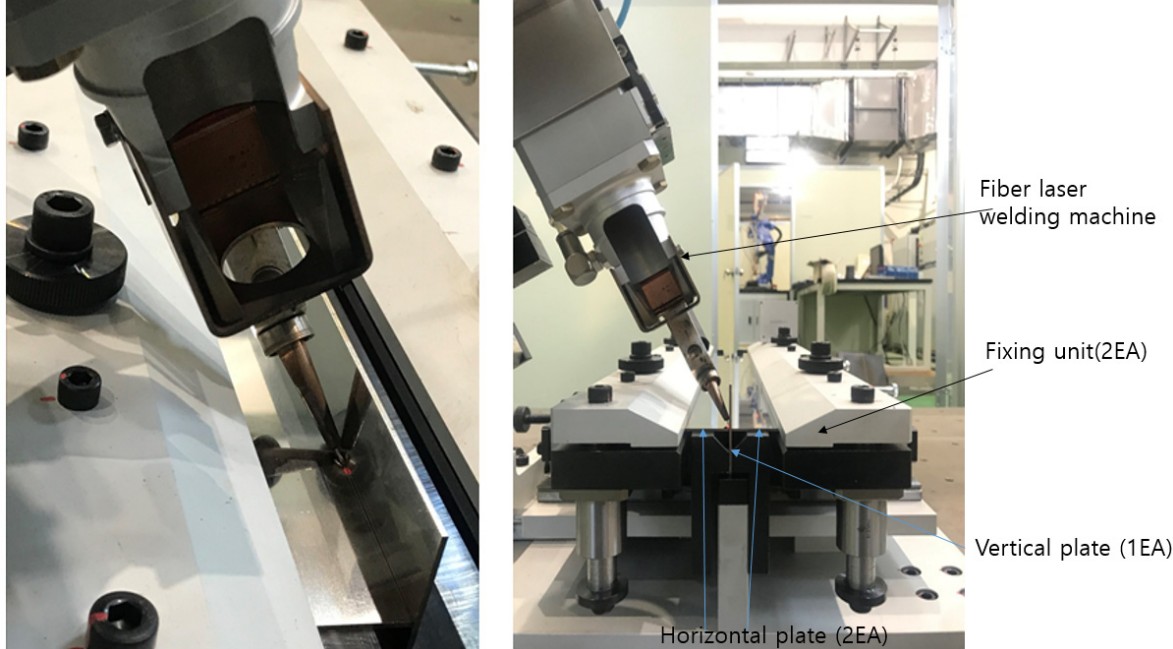

**Figure 2.** Shape of laser welding jigs.

## 2.2. Laser Welding Using Fiber Laser System

Miyachi's 5 kW laser welding using fiber laser equipment (Miyachi, Chiba, Japan) was used to perform the laser welding and initial temperature was set to 0 °C. The equipment consists of a laser welding oscillator, an optical system, a controller, and a chiller, as shown in Figure 3. The optical system used in this study has a spot diameter of 400 μm, a focal length of 148.8 mm, and a focal depth of 6 mm. The wire feeder was manufactured by Powwel (Seoul, Korea) and the wire feeding velocity was min. 0.3 and max. 5 m/min. The 6-axis automatic robot was from Yaskawa (Kitakyushu, Japan) and makes possible the constant focus position of laser welding. The shielding gas was 99.99% argon with 15 L/min speed.

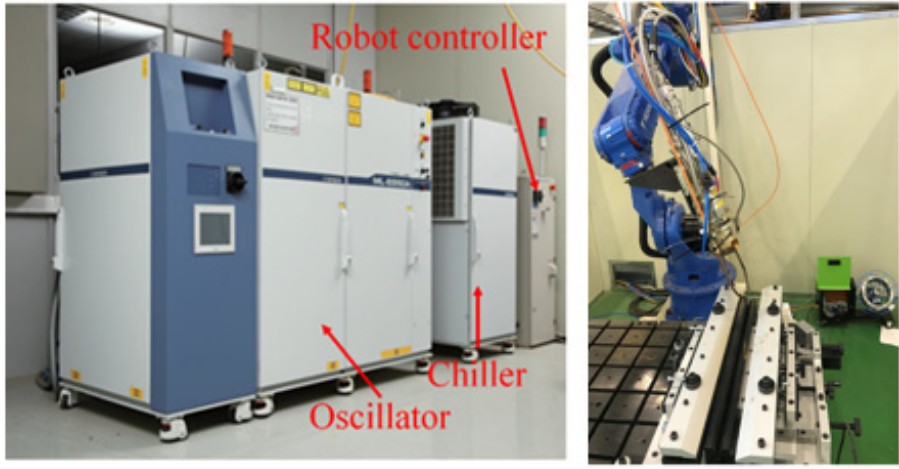

**Figure 3.** Fiber laser equipment.

## 2.3. Examination of Cross-Section

Cross-sectional examination was performed by cutting the center portion of a specimen to 10 (welding direction) × 25 mm (width direction). Polishing was performed to clearly examine the

shape of beads, and etching was performed with a Nital solution (a mixture of 10% HNO$_3$ and ethanol). The etched cross-section was examined using an optical microscope (Olympus), as shown in Figure 4.

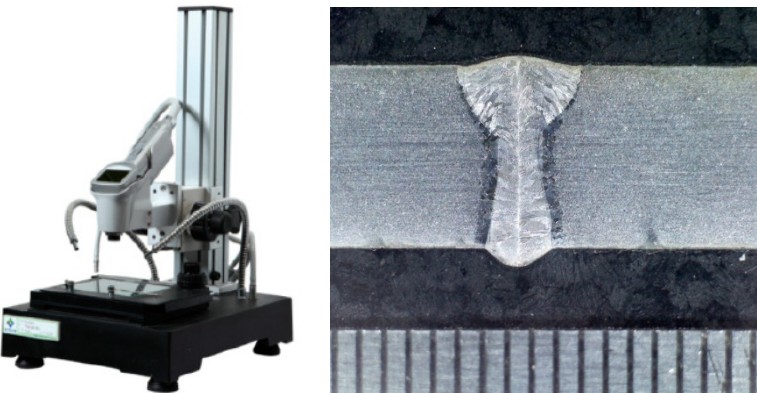

**Figure 4.** Optical microscope equipment and example of cross-section examination.

*2.4. Experimental Conditions of Cold Wire Laser Welding Using Fiber Laser*

Experiments were performed while changing experimental variables, i.e., laser power, wire feed rate, working angle, and beam location. The welding speed was set to 1 m/min, the gap between the base materials was set to 0 mm, and the focal length was the same in all experiments. Definitions of the related terms are shown in Figure 5.

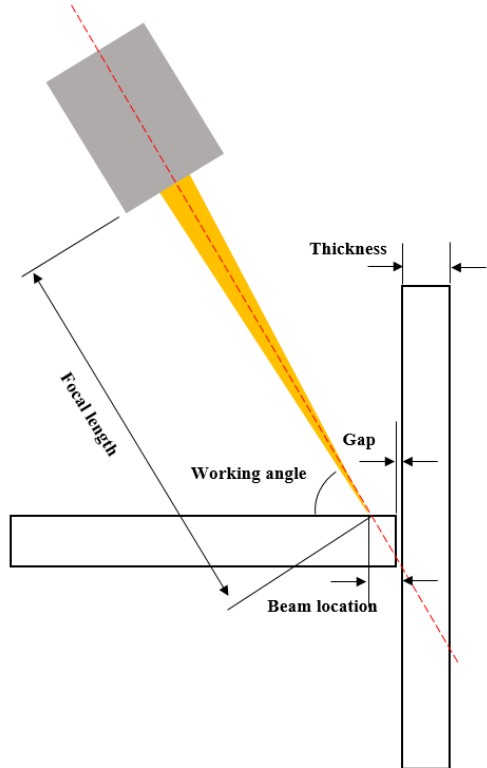

**Figure 5.** Definition of experiment laser welding conditions. (Focal length: Distance from the center of the lens to the focal point; Working angle: Angle between the welding unit and a horizontal plate; Beam location: Distance from focal point to the corner between the horizontal plate and vertical plate; Gap: Distance from vertical plate to the edge of the horizontal plate).

　　　The experimental conditions of laser welding for Invar with 1.5 mm thickness are shown in Table 3, while the conditions for 3.0 mm Invar are shown in Table 4. To find the optimal welding conditions to secure weldability, experiments were performed using a trial and error method.

**Table 3.** Conditions of laser welding for Invar with 1.5 mm thickness.

| Variables | Cases |
|---|---|
| Laser Power (kW) | 1.25, 1.75, 2.00, 2.25 |
| Wire Feeding Speed (m/min) | 1.00, 1.25, 1.50, 1.75 |
| Working Angle (°) | 60, 70, 80 |
| Welding Speed (m/min) | 1.0 |
| Beam Location (mm) | 0 |

**Table 4.** Conditions of laser welding for Invar with 3.0 mm thickness.

| Variables | Cases |
|---|---|
| Laser Power (kW) | 1.75, 2.25, 2.75, 3.25 |
| Wire Feeding Speed (m/min) | 1.75, 2.00, 2.25, 2.50, 3.00, 3.50 |
| Working Angle (°) | 70, 75, 80, 85 |
| Welding Speed (m/min) | 1.0 |
| Beam Location (mm) | 0.25, 0.5, 1.0 |

　　　The wire 0.8 mm used in this study is Invar M93, which is same composition of base material, FeNi36, made by Aperam in France. The wire feeding direction was horizontal to the welding direction and perpendicular to the laser welding beam.

　　　There are four experimental variables: laser power, wire feeding speed, working angle, and beam location and the value of welding speed fixed by 1 m/min. When four cases are tested in full factorial, 256 experiments must be performed. However, this is an inefficient and low-precision method. In this study, initial values were set based on the researchers' insights, and the conditions that met welding feasibility were explored through a trial and error method. Although it is necessary to secure the weldability of a cross-shaped specimen by fillet welding in two fillet welds, the welding test and evaluation for one side specimen were performed to secure the optimal conditions when searching for welding conditions because the structure is symmetrical. Then, the conditions were finally applied to the cross-shaped structure through welding at both sides.

## 3. Results

### 3.1. 1.5 mm Thickness of Invar Experiment

　　　The initial conditions for the 1.5 mm thickness Invar experiment were set as shown in Table 5. The welding cross-sections depending on the change inworking angle were examined under the initial conditions, and the cross-sectional shapes are shown in Figure 6.

**Table 5.** Welding conditions of Step 1 (working angle change, 3 cases).

| Variable | Value | Variable | Value |
|---|---|---|---|
| Laser Power (kW) | 1.25 | Working Angle (°) | 60, 70, 80 |
| Wire Feeding Speed (m/min) | 1.0 | Beam Location (mm) | 0.0 |

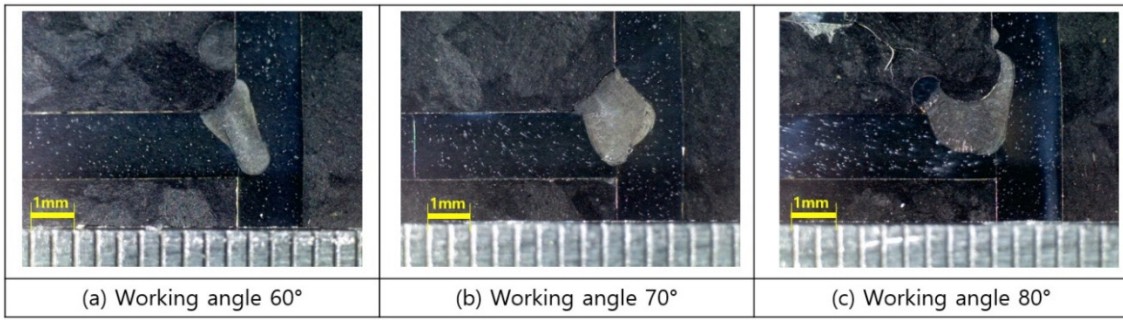

**Figure 6.** Difference among working angles.

As shown in Figure 6, there is no sufficient penetration to the opposite corner under the conditions described in Step 1 when examining the welding cross-sections. To solve this problem, it was determined that a higher laser power was necessary, so the laser power was increased to 1.75, 2.0, and 2.25 kW, from the 1.25 W used in Step 1. Regarding working angle, there was a welding defect, i.e., undercut, when the angle was 80°, so it was changed to 70° for the next experiment. The experimental conditions of Step 2 are described in Table 6 and the cross-sectional examination results are shown in Figure 7.

**Table 6.** Welding conditions of Step 2 (laser power change, 3 cases).

| Variable | Value | Variable | Value |
|---|---|---|---|
| Laser Power (kW) | 1.75, 2.0, 2.25 | Working Angle (°) | 70 |
| Wire Feeding Speed (m/min) | 1.0 | Beam Location (mm) | 0.0 |

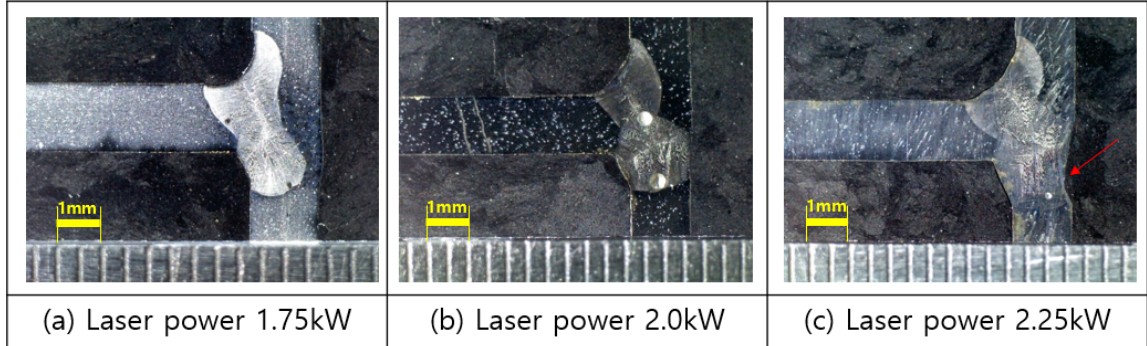

**Figure 7.** Differences among laser powers.

As shown in Figure 7, there was sufficient penetration in Step 2 but there were also undercuts under three laser power conditions. In addition, there was unexpected welding distortion for the 2.25 kW case in the opposite edge of the welding area in vertical plate. Therefore, the laser power was limited to 1.75 and 2.0 kW in Step 3. To address the welding defect, the wire feeding speed was increased to 1.0, 1.25, 1.50, and 1.75 m/min from 1.0 m/min (Step 1) in Step 3 and the result was examined. The experimental conditions of Step 3 are described in Table 7 and the cross-sectional examination results are shown in Figures 8 and 9.

**Table 7.** Welding conditions of Step 3 (laser power change, wire feeding speed change, 6 cases).

| Variable | Value | Variable | Value |
|---|---|---|---|
| Laser Power (kW) | 1.75, 2.0 | Working Angle (°) | 70 |
| Wire Feeding Speed (m/min) | 1.0, 1.25, 1.50, 1.75 | Beam Location (mm) | 0.0 |

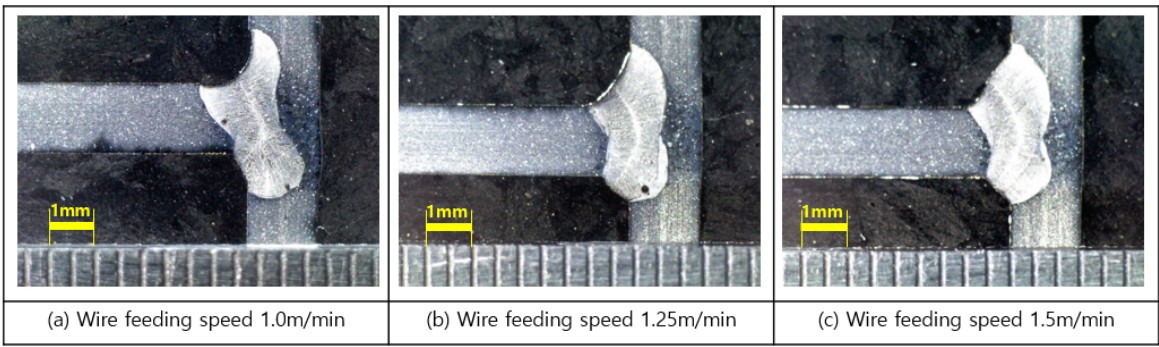

**Figure 8.** Different wire feeding speeds with 1.75 kW of laser power.

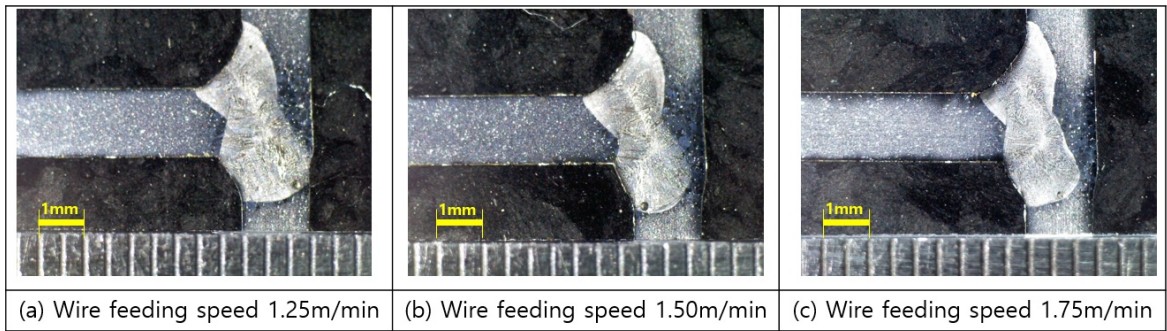

**Figure 9.** Different wire feeding speeds with 2.0 kW of laser power.

As shown in Figures 8 and 9, the most advantageous conditions considering welding defects at the upper bead and the bottom bead are laser power of 1.75 kW and wire feeding speed of 1.5 m/min. Using these conditions as best trial conditions, two fillet welds of double-sided welding for the cross-shaped structure of a 1.0 mm-thick Invar material were performed twice, and the cross-sectional examination results are shown in Figure 10.

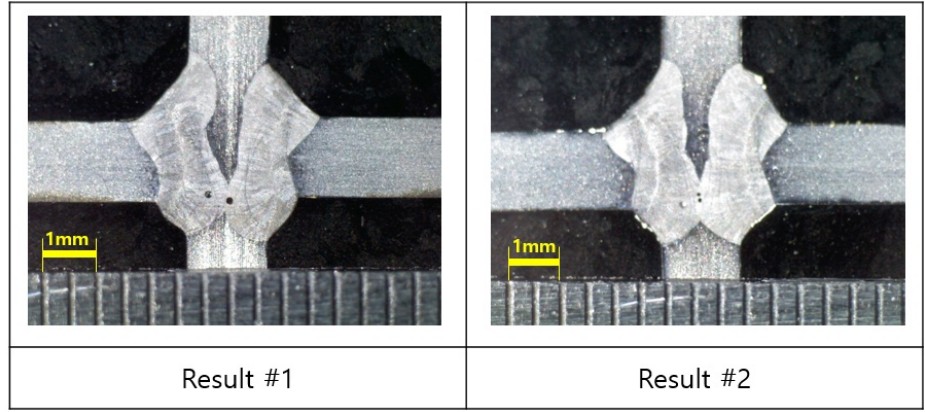

**Figure 10.** Final result of 1.5 mm thickness Invar.

### 3.2. 3.0 mm Thickness of Invar Experiment

As with the 1.5 mm thick Invar, a trial and error method was applied for the 3.0 mm thick Invar. In Step 1, an experiment to determine the range of initial welding conditions was performed using laser power as a variable. The detailed welding conditions are described in Table 8 and the cross-sectional examination result for each experimental condition is shown in Figure 11.

**Table 8.** Welding conditions of Step 1 (laser power change, 4 cases).

| Variable | Value | Variable | Value |
|---|---|---|---|
| Laser Power (kW) | 1.75, 2.25, 2.75, 3.25 | Working Angle (°) | 70 |
| Wire Feeding Speed (m/min) | 1.5 | Beam Location (mm) | 0.0 |

**Figure 11.** Applying different levels of laser power.

As shown in Figure 11, there was not sufficient penetration for fillet welding of 3.0 mm thick Invar material when the welding power was 2.75 kW or less. Therefore, in Step 2, the experiment was performed by applying laser power higher than 2.75 kW, at 3.0, 3.25, and 3.5 kW, and the beam location was changed from 0 to 0.5, 1.0 mm. The experimental conditions are shown in Table 9 and experimental results are shown in Figures 12 and 13.

**Table 9.** Welding conditions of Step 2 (laser power change, beam location change, 6 cases).

| Variable | Value | Variable | Value |
|---|---|---|---|
| Laser Power (kW) | 3.0, 3.25, 3.5 | Working Angle (°) | 70 |
| Wire Feeding Speed (m/min) | 1.5 | Beam Location (mm) | 0.5, 1.0 |

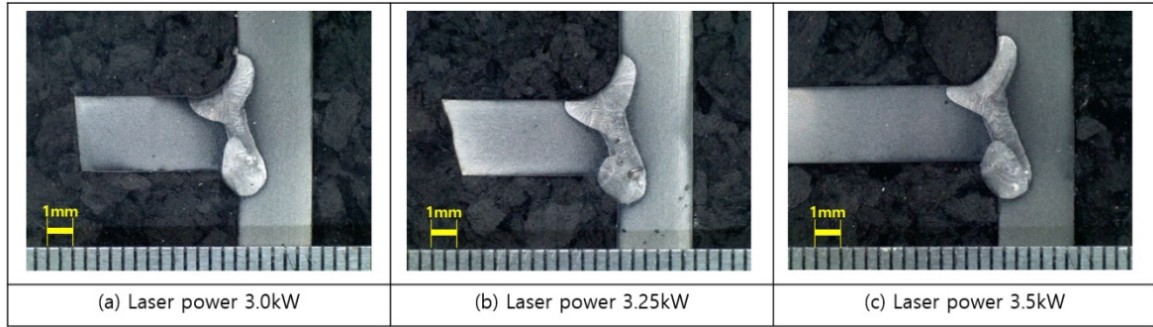

**Figure 12.** Applying different levels of laser power with 0.5 mm beam location.

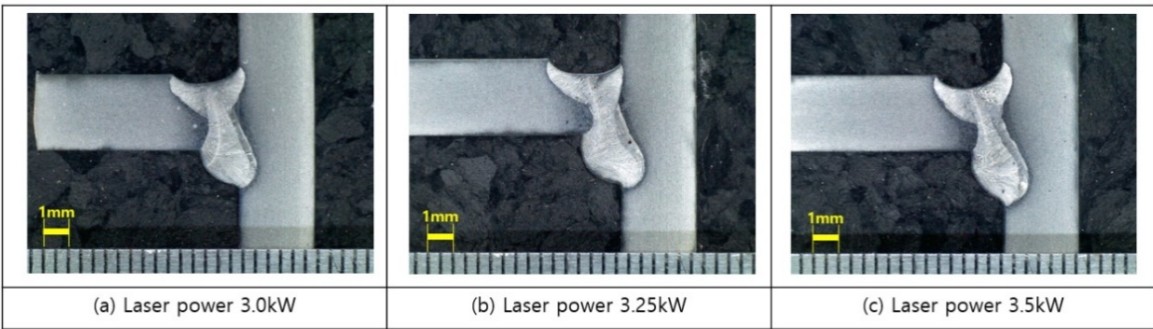

**Figure 13.** Applying different levels of laser power with 1.0 mm beam location.

As shown in Figure 12, the fillet welding area was not sufficiently filled when the beam location was 0.5 mm. In addition, as shown in Figure 13, there was an undercut and there was not proper welding when the beam location was1.0 mm. To solve this problem, in Step 3, the wire feeding speed was increased from 1.5 m/min. The experimental conditions are shown in Table 10 and the experimental results are shown in Figures 14 and 15. At this time, the beam location was set to 0.25 and 0.5 mm, respectively, to examine the effect of beam location.

**Table 10.** Welding conditions of Step 3 (wire feeding speed change, beam location change, 12 cases).

| Variable | Value | Variable | Value |
| --- | --- | --- | --- |
| Laser Power (kW) | 3.0 | Working Angle (°) | 70 |
| Wire Feeding Speed (m/min) | 1.75, 2.0, 2.25, 2.5, 3.0, 3.5 | Beam Location (mm) | 0.25, 0.5 |

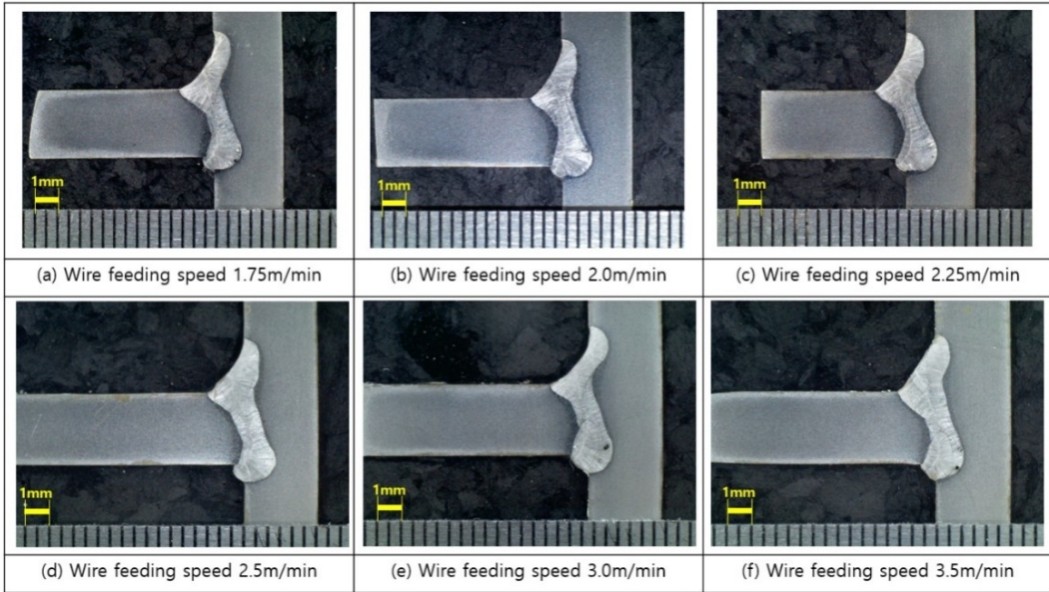

**Figure 14.** Different wire feeding speeds with 0.25 mm beam location.

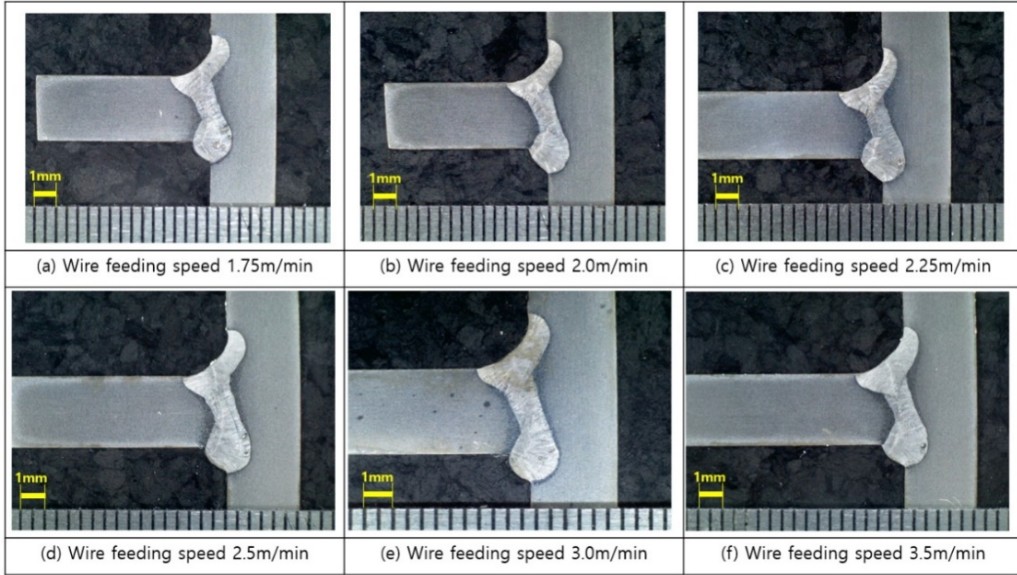

**Figure 15.** Different wire feeding speeds with 0.5 mm beam location.

As shown in Figures 14 and 15, there is no welding defect in Step 3 and the most advantageous conditions for the upper and bottom beads are beam location of 0.25 mm and wire feeding speed of 3.5 mm/min. Using these conditions as optimal conditions, two fillet welds of double-sided welding for cross-shaped structure of a 3.0 mm thick Invar material were performed twice, and the results of the cross-sectional examination are shown in Figure 16.

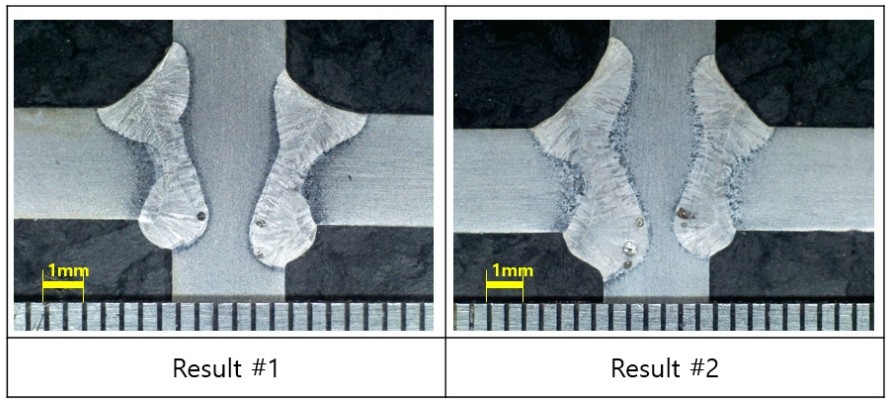

**Figure 16.** Final result of 3.0 mm thickness Invar.

## 4. Discussion

In this study, research on the feasibility of fillet welding for an Invar structure through laser welding using fiber laser with cold wire was performed. To find proper welding conditions, four variables (laser power, working angle, wire feeding speed, and beam location) were considered. Laser power had a decisive effect on the welding depth, which is shown in Figure 11. There was not enough welding depth with the condition below 2.75 W of laser power for 3.0 mm thickness Invar. However, the high laser power can make undercut as Figure 13, so appropriate laser power was very important for welding feasibility. Other variables such as working angle, wire feeding speed, and beam location determined the dimension of welding fillets. They affected the welding fillet not only at the corner beneath the cold wire but at the opposite corner. As the variables and consequences did not display a consistent relationship, searching the appropriate condition was completed with the complex conditions of four variables.

Although at least four fillet welds are required when fabricating a cross-shaped structure through typical TIG welding, only two welding passes are required for the method proposed in this study, thus reducing processing and equipment setting time. In addition, fabrication time is reduced (Table 11) because the speed of laser welding using fiber laser is two times faster than that of general TIG welding (Table 11). In addition, it is better than general TIG welding in terms of reducing welding distortion, because laser welding using a fiber laser, which is advantageous in terms of having less welding distortion, is used.

**Table 11.** Estimated effectiveness in comparison with TIG welding.

| Welding Type | Welding Speed (m/min) | Number of Welding | Welding Time (s) (5 m Length Structure) |
|---|---|---|---|
| Fiber Laser Fillet Welding | 1.0 | 2 | 600 |
| TIG Welding | 0.5 | 4 | 2400 |

To verify the welding qualities, a test of internal defects in the welding areas such as an ultrasonic test or X-ray test should be performed, as well as verification through tests of mechanical properties such as hardness, strength, fatigue, etc. This study examined the feasibility of a welding method that reduces the number of fillet welds using a fiber laser; further study in Part II should include a test of internal defects and mechanical properties.

## 5. Conclusions

This study is about fillet welding using a fiber laser and cold wire feeding when fabricating a cross-shaped Invar structure in an LNG storage tank. Laser welding using a fiber laser has a high welding speed and little welding distortion, giving it advantages when fabricating a structure requiring high precision. With the cold wire feeding, it was possible to compensate for the undercut and the loss of the base metal when laser welding was performed alone, and a good welding bead shape could be obtained. The main conclusions of this work are the following:

(1) When fabricating a cross-shaped structure through general TIG welding, fillet welding should be performed for each of the four corners with PJP. However, fillet welding is performed for two corners only to reduce the number of weld passes if the fiber laser fillet welding method proposed in this study is used because of the deep keyhole of CJP. In addition, the overall welding time can be reduced to less than one quarter of the time required for TIG welding, because laser welding using fiber laser speed is two times faster.

(2) Using laser power, wire feeding speed, working angle, and beam location as variables, the welding conditions for two corner fillet welding were determined. At this time, welding was performed using a trial and error method, in which each variable was changed and its feasibility was verified through cross-section examination.

(3) To fabricate a cross-shaped Invar structure with 1.5 mm thickness, welding feasibility was achieved under the conditions of laser power of 1.75 kW, wire feeding speed of 1.5 m/min, working angle of 70°, and beam location of 0 mm. For the 3.0 mm Invar structure, welding feasibility was achieved under the conditions of laser power of 3.0 kW, wire feeding speed of 3.5 m/min, working angle of 70°, and beam location of 0.5 mm. Although those conditions are the best trials of this research, there could be other conditions for the feasibility for weldability.

(4) This study is a preliminary study for improving the fabrication method of cross-shaped Invar structures using fiber laser fillet welding. Subsequent studies should aim to verify the soundness of such welded structures through tests of mechanical properties such as hardness, strength, and fatigue, in addition to checking for welding defects using ultrasonic and X-ray testing. Moreover, for obtaining the metallurgical soundness, a follow-up study will be performed including finding the condition for the removal of the porosity after welding.

**Author Contributions:** Conceptualization: D.-S.K.; C.P.; J.K. (Jaewoong Kim); J.K. (Jisun Kim), experiment: D.-S.K.; J.K. (Jisun Kim); H.-K.L., software: C.P., validation: D.-S.K.; H.-K.L.; C.P.; J.K. (Jaewoong Kim); J.K. (Jisun Kim), paper research: C.P.; J.K. (Jaewoong Kim); J.K. (Jisun Kim); H.-K.L., data analysis: C.P.; J.K. (Jaewoong Kim), writing—original draft preparation: C.P., writing—review and editing: D.-S.K.;J.K. (Jaewoong Kim); J.K. (Jisun Kim), supervision: J.K. (Jaewoong Kim), project administration: J.K. (Jaewoong Kim). All authors have read and agreed to the published version of the manuscript.

**Funding:** This study was conducted with the support of the Korea Institute of Industrial Technology as "Study on the Core Element Technology for Smart Mobility (PJA20073)".

**Conflicts of Interest:** The authors declare no conflict of interest.

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
