# Peer review of "A Study on Cross-Shaped Structure of Invar Material Using Cold Wire Laser Fillet Welding (PART I: Feasibility Study for Weldability)"

_metals, doi:10.3390/met10101385_

Round 1
Reviewer 1 Report
The topic of the article is relevant. The methodology of the experiments is appropriate and sufficient. Unfortunately the scientific level is relative low and thus the article doesn’t provide sufficient comprehension. But this can be improved applying appropriate statistical analysis of the presented data. In this sense, the denotation “optimum” (Section 3.1 and 3.2) is wrong, it is rather the “best trial”. Furthermore, the evaluation criteria must be clearly defined. Accordingly, the conclusion “3” is not really soundly justified.
Critical is also that some citations are incorrect or insufficient:
Line 48: “OLED display industry [7,8]” - I can’t see any relation to the topic of the article, nether in the physics nor in the problem scale. Please, remove or explain it precisely.
Line 65-67: the references 15-17 deal neither with the considered welding methods nor with the welding distortions and residual stresses.
Line 70-73: the references 20-21 are not related to the statements.
I believe, these are accidentally exchanged reference records, please check.
Further, the following minor changes are suggested:
Line 37: the abbreviation “LSF” need to be denoted
Line 54-57: the figures are not really informative, it is not clear, what the authors want to illustrate. Shell they indicate the position and kind of the welded joints, subject of the investigation?
Line 67, 76, 237 and may be other: the term “deformation” is used incorrect. Deformations occur at every welding process. In the common case their magnitude depends on the material, not on the process, i.e., method. But the domain where they act, i.e., the width of the deformed region, depends on the method. Thus, it results to alternation, e.g. of the distortion.
Line 71, 80, 85, 101, 109, 126, 252, 254, 265 and may be more: “fiber laser welding”!? the welding methods are classified! So, known are “fiber laser” as kind of laser system, and “laser welding” as welding method. For example: laser welding using fiber laser.
Line 74: please revise … or “# (number shaped)” …
Line 80, 82, 85, 185, 231, 233, 242 etc.: the classification of the welds or the term “welding pass” is used imprecise, for example “four welding passes” instead “four fillet welds”, etc., otherwise please clarify.6,
Section 2.3: Could you please provide also information about the laser caustic, focusing, etc. as well as the initial temperatures
Line 153, 190: please check the subtitles
Line 210-211: It is not clear, what the authors mean.
Author Response
Thank you for your valuable comments. Please check the attached file.

Reviewer 2 Report
This manuscript presents an experimental investigation on the laser welding of INVAR material. The research findings of this manuscript could be useful. However, the current manuscript should be improved:
- The used "trial & error" approach does not allow to find the real optimum of the welding parameters and conditions. The authors should better describe the results as "appropriate" or "suitable".
- Table 1: Please check the unit of the coefficient of thermal expansion (µm/mmK). Please provide the references for the values in the table and for which temperatures they are valid.
- Figure 3: The meaning of the picture is not clear. Please add a description.
- Introduction: The introduction focusses mainly the used material INVAR for LNG facilities. But the scientific review of literature about welding of INVAR material is missing. Please provide an international literature review about welding, particular laser welding, of INVAR materials and comparable cross-shaped structures.
- Introduction: Laser welding with filler materials is cited by references about Laser-Hybrid Welding [19-22]. Since this hybrid process differs from laser welding with cold wire, references should be appropriate updated.
- Table 2: The “Balance” should be added.
- Figure 6 and 7: The pictures only show devices and do not contribute to the content of the manuscript. They can be omitted.
- Experiments: The description about the edge preparation of the plates resp. information about realization of the gap of 0 mm should be added. Furthermore, there are no information about the filler wire and its feeding (diameter, material, feeding direction with respect to the welding direction). Which shielding gas was used? Which focus position was used in the experiments and was it constant or has been changed.
- Subtitles 3.1 and 3.2: Probably a misspelling, 5 mm should be changed to 1.5 mm and 0 mm to 3 mm resp.?
- Discussion: The brief discussion is more a summary of the benefits of INVAR laser beam welding than a scientific justification of the results. The authors should add further discussion of the experimental results. The discussion should explain the relationship between welding parameters and conditions, heat input and the resulting weld shape. The results obtained can be compared with results from references.
- Discussion: The statement that laser beam welding is better than TIG welding in terms of distortion is not proven in the paper (page 12, line 235).
Author Response

(The authors gave the same response as above.)

Reviewer 3 Report
This is a very good experimental work, the objective of this work was to optimize the parameters for a geometrically sound weld and it has been reached. However you are showing macrographs of welds containing some porosity and you don't mention that. I understand that your plan is to write a second paper with in-depth study of the metallurgical soundness of the welds but this is somewhat frustrating for the reader. Optimization should take into account both geometrical and metallurgical soudness.
This is why your paper is not as interesting as it should be although it is impressive to see those joints with almost no distorsion.
You should correct :
- subtitle 3.1 : 1.5mm instead of 5 mm
- subtitle 3.2 : 3 mm instead of 3 mm
Author Response

(The authors gave the same response as above.)

Round 2
Reviewer 1 Report
Dear Authors,
I regret to see that only the minor corrections have been accepted.
Their still no sufficient analysis of the experimental results and thus, it limits the soundness of your work significantly. I agree your pointer that these specimens are subject of further investigations but in this stage you have just an observation without proof of its reliability and consistency. So, unfortunately, I have some objections to the scientific quality of your article.
Considering the statement: "Laser welding applies a concentrated heat source to the small area of a material’s surface within a short period of time, so it has relatively small welding distortion compared to other welding methods" (formerly Line 65-67). Neither 14 nor 15 deals with welding distortions, moreover neither from them compares LBW with TIG or any arc welding. The statement itself is correct and no doubt widely accepted. It is ethical question, maybe it is better to remove the references rather than give inappropriate ones.
Author Response
First of all, thank you again for your high opinion, and I generally agree. We will reflect your opinions in further research and do better research. In response to your opinion, we have deleted the sentence for Ref[14-15].Reviewer 2 Report
Thank you for improving the manuscript. The new introduced reference [18] in the reference list is not complete. Please correct it.
Author Response
Thank you for your opinion.
And I revise it and add the doi of ref[18].